# Wings and halteres act as coupled dual oscillators in flies

Tanvi Deora[1], Siddharth S Sane[2], Sanjay P Sane[3]*

[1]Department of Biology, University of Washington, Seattle, Washington, United States; [2]Department of Physics, Azim Premji University, Bangalore, India; [3]National Centre for Biological Sciences, Tata Institute of Fundamental Research, Bangalore, India

**Abstract** The mechanics of Dipteran thorax is dictated by a network of exoskeletal linkages that, when deformed by the flight muscles, generate coordinated wing movements. In Diptera, the forewings power flight, whereas the hindwings have evolved into specialized structures called halteres, which provide rapid mechanosensory feedback for flight stabilization. Although actuated by independent muscles, wing and haltere motion is precisely phase-coordinated at high frequencies. Because wingbeat frequency is a product of wing-thorax resonance, any wear-and-tear of wings or thorax should impair flight ability. How robust is the Dipteran flight system against such perturbations? Here, we show that wings and halteres are independently driven, coupled oscillators. We systematically reduced the wing length in flies and observed how wing-haltere synchronization was affected. The wing-wing system is a strongly coupled oscillator, whereas the wing-haltere system is weakly coupled through mechanical linkages that synchronize phase and frequency. Wing-haltere link acts in a unidirectional manner; altering wingbeat frequency affects haltere frequency, but not vice versa. Exoskeletal linkages are thus key morphological features of the Dipteran thorax that ensure wing-haltere synchrony, despite severe wing damage.

## Editor's evaluation

This manuscript examines how the mechanical linkages in the thorax of flies help these animals maintain symmetric wing motion in the face of uni- or bilateral wing damage. In previous work, the authors showed that these same linkages play an important role in maintaining the proper relative phase relationship between the wing and the haltere, a multifunctional sensory unique to flies. Through delicate manipulations of the thorax, wing, and haltere, the authors' experimental results support a mechanical model of the thorax they previously proposed known as the coupled dual-oscillator hypothesis, where mechanical linkages in the thorax both enable symmetric wing motion as well as coordinate haltere oscillation relative to the wing.

*For correspondence: sane@ncbs.res.in

**Competing interest:** The authors declare that no competing interests exist.

## Introduction

Flies are among the best exemplars of aerial agility. The Dipteran order encompasses a vast repertoire of flight types ranging from the exquisite hovering and maneuvering ability of hoverflies, to the stable trajectories of mosquitoes and rapid territorial chases in houseflies (*Land and Collett, 1974*). Such complex maneuvers require precise and rapid control, guided by sensory feedback from multiple modalities (*Bender and Dickinson, 2006*; *Heide and Götz, 1996*; *Hengstenberg, 1993*; *Pringle, 1997*; *Sherman and Dickinson, 2003*; *Trimarchi and Schneiderman, 1995*). Of particular importance for flight stability is the mechanosensory feedback from halteres – the modified hindwings of flies – which sense gyroscopic forces during aerial maneuvers (*Nalbach, 1994*; *Nalbach, 1993*; *Nalbach*

and Hengstenberg, 1994; Pringle, 1997). During flight, the halteres oscillate in a constant plane at frequencies that are identical to their flapping wings, and with a constant phase difference relative to the wings. During an aerial turn, an externally imposed change in the plane of haltere oscillation is resisted due to rotational inertia, causing Coriolis torques to act on the haltere base. Mechanical strain in the haltere shaft due to Coriolis torques is sensed by multiple fields of campaniform sensillae distributed around its base. These encode the stroke-by-stroke status of aerial rotations and provide mechanosensory feedback to the wing muscles (Fayyazuddin and Dickinson, 1996; Yarger and Fox, 2018). In addition to detecting body rotations, haltere plays an additional role in mediating visually driven maneuvers; visual feedback drives haltere-steering muscles and alters the haltere mechanosensory feedback (Chan and Dickinson, 1996; Dickerson et al., 2019). Importantly, the relative phase difference between the feedback from wing and haltere mechanosensors determines the activity patterns in wing-steering muscles (Fayyazuddin and Dickinson, 1999; Fox et al., 2010). During flight, the two wings of flies move exactly in-phase relative to each other, whereas halteres move at a constant phase offset relative to wings. This precise phase coordination is maintained at wingbeat frequencies that far exceed 100 Hz (Deora et al., 2015; Hall et al., 2015). Because even slight asymmetries in the bilateral wing motions can result in significantly large torques on the body during flight (Fry et al., 2003), the phase and frequency synchronization of the wing-haltere kinematics is a core feature of Dipteran flight, any deviation from which may signal either a self-generated aerial turn or an unwanted perturbation.

Previous research has shed much light on the architecture of the Dipteran thorax (Boettiger and Furshpan, 1952; Deora et al., 2017; Ennos, 1987; Miyan and Ewing, 1997; Pringle, 1949; Walker et al., 2014). The wing-haltere system acts as a complex resonant box. In flies, wings are actuated by two sets of antagonistic indirect flight muscles aligned dorso-longitudinally and dorso-ventrally within the thorax (Deora et al., 2017; Dickinson and Tu, 1997; Pringle, 1949). These muscles do not directly articulate at the wing base, but instead attach to the thorax, thereby indirectly powering the wing motions. Their activation is myogenic; hence, contraction in one set of muscles triggers delayed contraction of the other and vice versa, setting up resonant cycles of oscillations of the entire thorax (e.g., Deora et al., 2017). A complex wing hinge transforms oscillatory deformations of the thorax into large-amplitude wing strokes. The indirect flight muscles require neural stimulation to remain in an active state, but the frequency of stimulation is typically an order of magnitude lower than resonant thoracic oscillations. The attitude of the wing is finely adjusted on a stroke-to-stroke basis by a set of steering muscles that are under direct neuronal control (Lindsay et al., 2017). Thus, the frequency characteristics of Dipteran wing movements are set by the resonance frequency, which in turn is determined primarily by the wing-thoracic morphology. Thus, any alteration in the wing length causes corresponding changes in their inertial and aerodynamic loads, and hence in the frequency of the wingbeat. Unlike wings, the motion of each haltere is powered by a single asynchronous muscle whose contractions activate its upstroke, whereas the downstroke is thought to be entirely passive (Chan et al., 1998; Pringle, 1949).

Although wingbeat frequency depends on the wing-thorax morphology, the precise coordination of wings and halteres is mediated by mechanical linkages within the thorax that ensure tight coupling of phase and frequency (Figure 1B, Deora et al., 2015). This suggests the hypothesis that wings and halteres act as independent forced oscillators, whose kinematics are both coupled and constrained by two separate mechanical linkages within the thorax (the coupled dual-oscillator hypothesis, Figure 1B). One linkage, the scutellar link (alternatively, wing-wing link, in orange in Figure 1B), is embedded within the scutellum and ensures that both wings are synchronized. A second linkage, the sub-epimeral ridge (alternatively, wing-haltere link, highlighted by green arrows in Figure 1B), ensures precise coordination of wings and halteres. This linkage ensures only weak coupling of the wing-haltere synchrony, but breaks down when wingbeat frequency exceeds a threshold value (Deora et al., 2015).

The resonant properties of such a system rely on the mechanical integrity of the wing-thorax system. However, wings of insects often undergo significant natural wear-and-tear during the lifetime of an adult insect (Hayes and Wall, 2002). Wing damage alters both frequency and aerodynamic force generation of the flapping wings (Hedenström et al., 2001; Kihlström et al., 2021; Muijres et al., 2017), thus posing a challenge to the overall coordination of wing motion. Such damage is typically asymmetric and can affect maneuverability. Not surprisingly, in insects such as bumblebees and

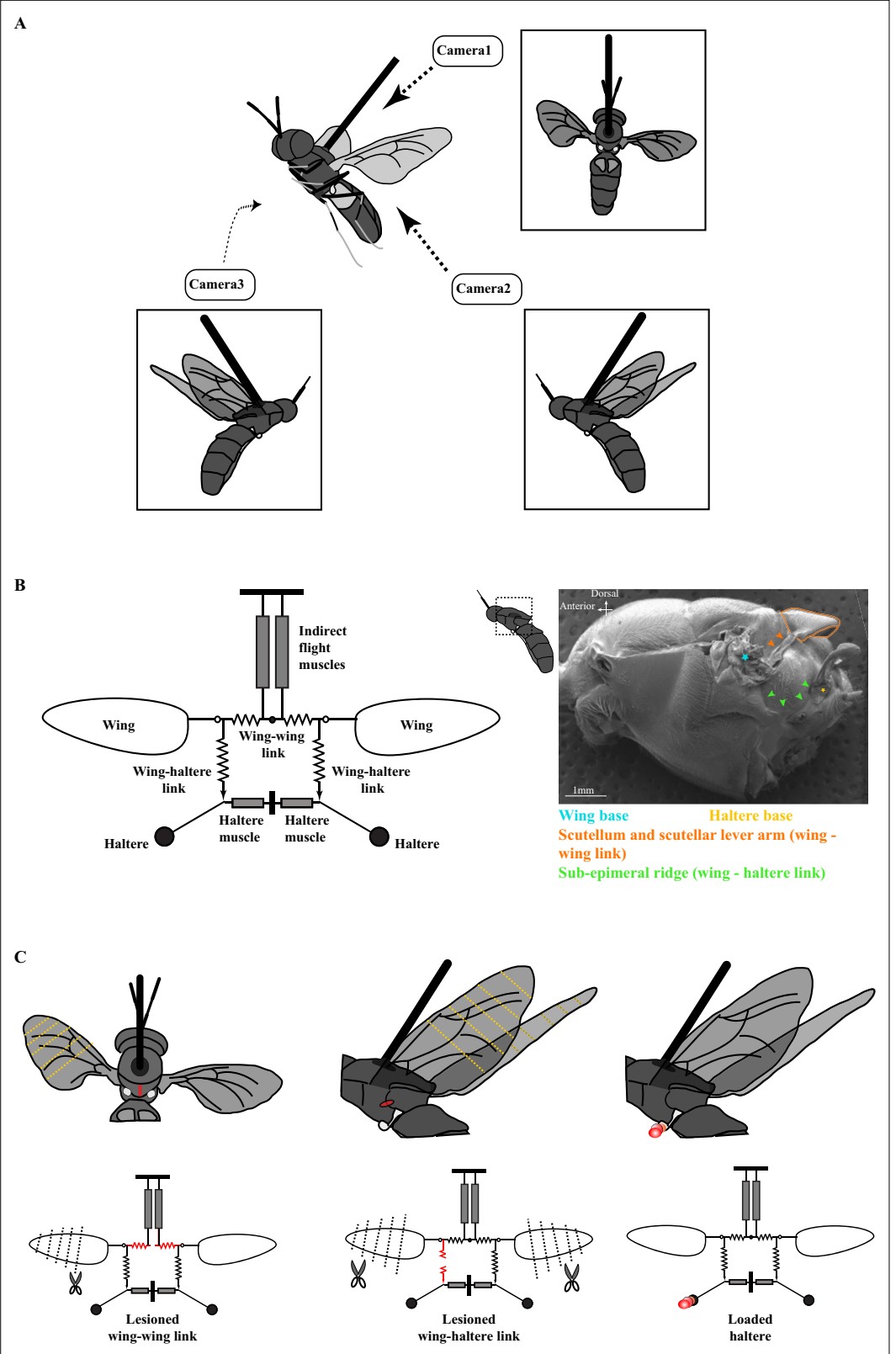

**Figure 1.** The experimental setup. (**A**) Schematic of a tethered fly showing the position of the three high-speed cameras. Insets show the three different camera views. (**B**) Mechanical model of the Dipteran thorax modified from an earlier work (left, *Deora et al., 2015*). This model excludes the clutch and gearbox from the previous figure (represented here by an open circle at the wing joint), focusing instead on the wing-wing and wing-haltere linkages

*Figure 1 continued on next page*

*Figure 1 continued*

that are the focus of the experiments described here. (Right) SEM image of the solder fly thorax in the lateral view highlighting the various linkages. (**C**) A schematic (top) and model (bottom) illustrating the experimental treatments. The treatments (red) included lesioning the scutellum or wing-wing link (left panel), lesioning sub-epimeral ridge or wing-haltere link (middle panel) and haltere loading (right panel). The clipped wings are also indicated with dotted lines. The same treatments are also shown in the model schematic and used as insets in later figures. *Figure 1B* has been adapted from **Figure 4G** from *Deora et al., 2015*.

---

dragonflies, wing damage leads to decreased success in hunting and also greater mortality (*Cartar, 1992*; *Combes et al., 2010*; *Haas and Cartar, 2008*).

Here, we address two related questions. First, how do flies maintain symmetric wing movement under conditions of wing damage? Second, in light of the coupled dual-oscillator hypothesis, how robust are the wing and haltere kinematics in the face of wing or thoracic damage? To address these questions, we conducted a series of experiments on the soldier fly, *Hermetia illucens*, in which we made specific lesions of scutellar linkages, and sub-epimeral ridge to impair the mechanical integrity of the thorax. In addition, we clipped the wings or loaded the halteres to alter their oscillation frequencies. These experiments enabled us to systematically test the predictions of the coupled dual-oscillator hypothesis (*Figure 1B*) and outline the key mechanical properties of the Dipteran thorax that ensure robust wing-haltere coordination.

## Results

### Asymmetric wing damage influences kinematics but does not alter wing coordination

The left and right wings are coupled in phase by a mechanical linkage running through the scutellum within the thorax such that the two wings always flap at constant phase relative to each other (*Figure 1B*; *Deora et al., 2015*). However, mechanical coupling of the two wings also implies that their flapping frequencies are identical. Clipping both wings symmetrically results in identical but elevated wingbeat frequencies (*Deora et al., 2015*). However, if the wings are mechanically coupled, altering the frequency of one wing should correspondingly alter the frequency of the contralateral wing. Thus, clipping the length of just one wing should result in intermediate increase in frequency – a combination of both the intact and shortened wing. Because the wing-wing link is intact, the two wings should flap at identical frequencies. To test these predictions, we filmed the wing motion of tethered soldier flies in which the length of only one wing was sequentially reduced while leaving the other intact. Wingbeat frequency of one-wing clipped fly was determined by sampling 20 arbitrarily chosen wingbeats. Even after drastic reduction in the length of one wing by >50% of the original length, the frequencies of the clipped and intact wings were always identical and did not drastically reduce from their intact values (*Figure 2A*; for additional data, see *Figure 2—figure supplement 1*, p=0.623; number of flies, n = 6, one-sided, Wilcoxon signed-rank test on the maximum wing difference). Thus, the overall wingbeat frequency is determined by the frequency of the intact wing.

### Scutellar integrity is essential for wing coordination

According to the coupled dual-oscillator hypothesis, lesioning the scutellum should decouple the frequencies of left and right wings. Hence, we severed the scutellar linkage and filmed the flies while again sequentially clipping one wing to reduce its aerodynamic resistance, thereby increasing its frequency. This resulted in irregular wingbeats in these flies, with frequent mid-stroke pauses and an overall reduction in stroke amplitude. Fourier analysis of the time series of wing motion shows that both wings oscillated at very different frequencies. Thus, unlike the intact scutellum case in which frequency synchronization was robust despite wing damage (*Figure 2A*), the wings of a scutellum-lesioned fly were decoupled from each other (*Figure 2B*, additional data in *Figure 2—figure supplement 2*, p=0.022; n = 5, one-sided Wilcoxon signed-rank test on the maximum wing difference). For example, in the typical case of a scutellum-lesioned fly (*Figure 2—figure supplement 2A*), the clipped wing flapped at 130 Hz when cut to 50% of its original length as compared with 85 Hz in the intact wing. Our previous work shows that lesioning the scutellum disrupts and reattaching completely restores the phase coordination between both wings (*Deora et al., 2015*); thus, scutellar integrity is

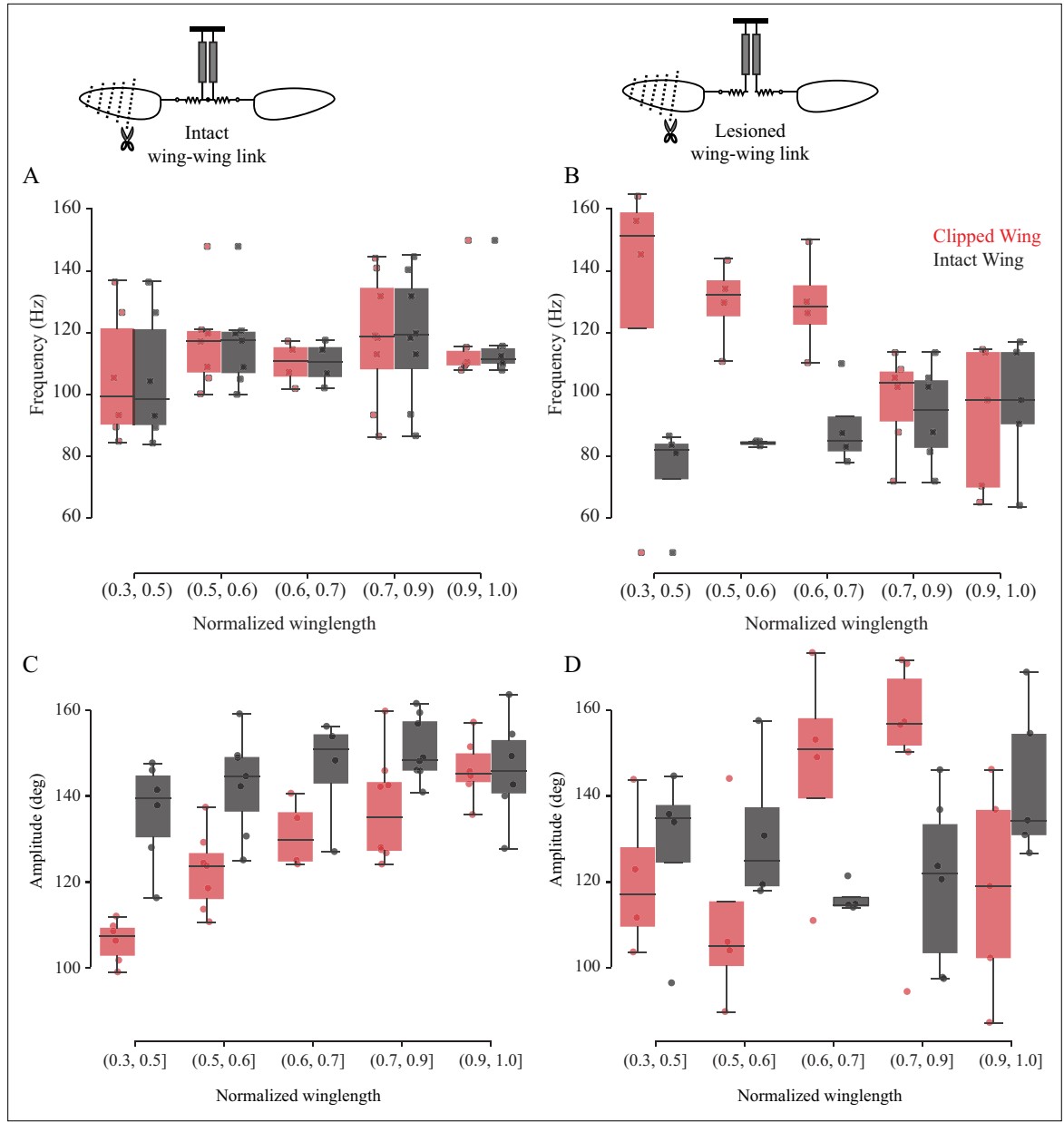

**Figure 2.** The frequency of the two wings is synchronized by the scutellum. Wingbeat frequency (**A, B**) and amplitude (**C, D**) of intact (gray boxes) and clipped (red boxes) wing as a function of clipped wing length for intact thorax (**A, C**) and slit scutellum (**B, D**). Insets show schematic for treatments. Each dot represents an individual fly. (**A**) Flies with intact thorax flap their wings at identical frequencies, whereas (**B**) the *scutellum-lesioned* flies flap at different frequencies. (**C**) In intact, tethered flies, the clipped wing moves through a smaller amplitude compared with the intact wing. (**D**) In scutellum-lesioned flies, both wings move erratically, with no consistent trend in wingbeat amplitude.

The online version of this article includes the following video and figure supplement(s) for figure 2:

**Figure supplement 1.** The frequencies of the two wings are coupled in intact flies.

**Figure supplement 2.** Scutellum synchronizes the frequencies of the two wings.

**Figure 2—video 1.** Left and top views of a soldier fly with left wing clipped and right wing intact (Figure 2).
https://elifesciences.org/articles/53824/figures#fig2video1

**Figure 2—video 2.** Right and top views of the soldier fly whose wing-wing link (scutellum) has been severed.
https://elifesciences.org/articles/53824/figures#fig2video2

essential to ensure precise coupling of both phase and frequency of the two wings, and imparts robust synchronization even if one or both wings are slightly damaged or torn. Together, these data show that the two wings are *strongly* coupled by the scutellar link.

## Amplitude control in response to wing damage

Wing damage not only alters the resonant frequency of the wing-thorax system but also reduces the aerodynamic force generation by the wings. Freely flying insects typically offset the effects of wing damage by altering their wing kinematics, particularly their wing amplitude (*Fernández et al., 2012*; *Kihlström et al., 2021*; *Muijres et al., 2017*; *Vance and Roberts, 2014*). To test the role of passive mechanics in altering wing kinematics, we quantified the amplitudes of the clipped and intact wings. In tethered flies with an intact thorax, the amplitude of clipped wing reduced as the wing was shortened, but there was no significant change in intact wing's amplitude(p=2.1e-4 for clipped, 0.218 for intact wings, Kruskal–Wallis H-test, *Figure 2C*). In contrast, neither the clipped nor the intact wings of tethered flies with a lesioned scutellum showed a significant difference in wing amplitude with reduction in wing length (p=0.067 and 0.184 for clipped and intact wings, respectively, Kruskal–Wallis H-test, *Figure 2D*).

## Sub-epimeral ridge weakly couples the frequency of each haltere to its ipsilateral wing

The wing and haltere motion on each side is coupled by a separate thoracic element called the sub-epimeral ridge (right panel, *Figure 1B*; *Deora et al., 2015*). Slight symmetrical clipping of both wings resulted in a small increase in wingbeat frequency and concomitant increase in haltere frequency. However, with further symmetrical reduction of wing length, wingbeat frequencies exceeded ~150% of the initial values, but the halteres failed to keep pace with the wings. In such conditions, haltere frequency dropped closer to their natural frequency, suggesting that their coupling was weak (*Figure 3*, control haltere [in blue]; *Deora et al., 2015*).

We next lesioned the sub-epimeral ridge on the left side while keeping the right side intact as internal control. If the sub-epimeral ridge is the main coupling link, then lesioning it should cause the haltere frequency on the lesioned (left) side to be decoupled from the increase in wingbeat frequency due to symmetrical wing shortening. Our data were consistent with this hypothesis; the control (right) haltere frequency matched the wing frequency more robustly than the lesioned left haltere-wing pair (*Figure 3A*, p=0.039 for wing-treatment haltere pair and p=0.657 for wing-control haltere pair, one-sided Wilcoxon signed-rank test at wing length bin = [0.6, 0.7], n = 6). Not surprisingly, these data were more variable. In four out of the six experiments, data were consistent with our hypothesis; the frequency of the haltere on the lesioned (left) side either did not increase at all (representative fly in *Figure 3B*, *Figure 3—figure supplement 1A and B*) or was decoupled from the wing even with slight changes in wing length, thus displaying no robustness in the wing-haltere synchrony (*Figure 3—figure supplement 1C*). In two flies, however, wingbeat frequency remained relatively unchanged despite clipping the wings incrementally, and haltere frequency on the lesioned side matched halteres on the control side (*Figure 3—figure supplement 1D and E*). Together, these results suggest that the sub-epimeral ridge weakly couples wing and haltere oscillation. Haltere motion can accommodate small to moderate changes in wingbeat frequency but fails if these changes are large.

## Integrity of the sub-epimeral ridge is essential for resonant oscillation of the thorax

In insects with an intact thorax, clipping the wings increases wingbeat frequency by as much as 90 Hz. In flies with unilaterally lesioned sub-epimeral ridge, the overall changes in wingbeat frequency (frequency of intact wing compared to the frequency at the shortest wing length) were relatively moderate (~60 Hz) even after shortening the wing to the smallest wing length (*Figure 4*, data for intact flies from *Deora et al., 2015*, p<0.05, Kruskal–Wallis ANOVA followed by the post hoc Tukey–Kramer multi-comparison test, n; eight intact flies, six flies for the other three treatments).

How does lesioning the sub-epimeral ridge alter wingbeat frequency, in addition to decoupling the wings and halteres? One possibility is that a lesioned ridge disrupts the anti-phase motion of wings and halteres, leading to aberrant haltere feedback to wing-steering muscles, thereby affecting

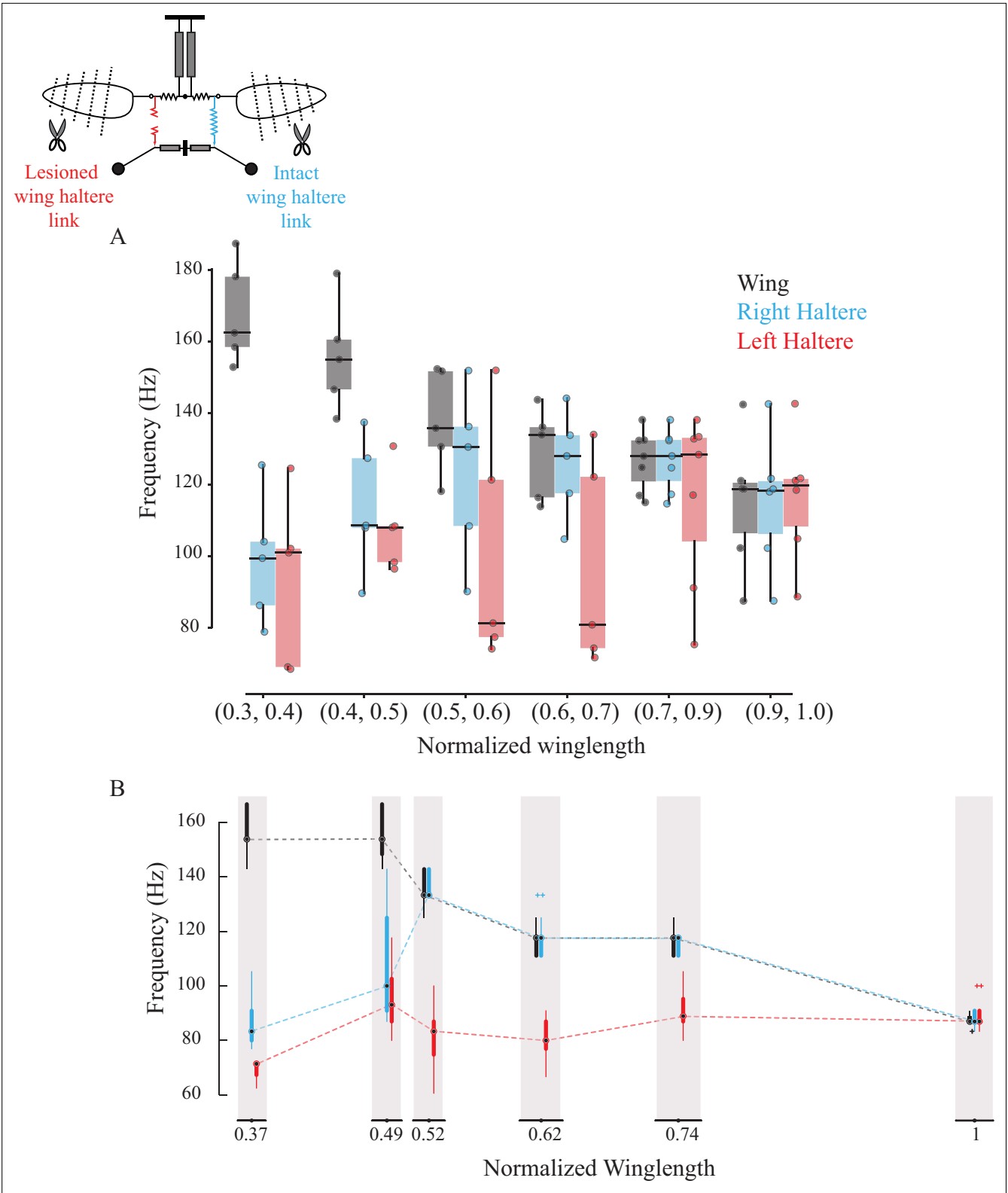

**Figure 3.** Sub-epimeral ridge couples the frequency of wings and halteres. Frequency of wing (gray), control haltere (blue), and haltere with the sub-epimeral ridge lesioned (red) as a function of wing length across all flies (**A**) and one representative fly (**B**). Inset shows the schematic for treatments. Each dot in (**A**) is an individual fly. Additional data for individual flies can be found in *Figure 3—figure supplement 1*.

The online version of this article includes the following video and figure supplement(s) for figure 3:

*Figure 3 continued on next page*

*Figure 3 continued*

**Figure supplement 1.** Sub-epimeral ridge couples the frequency of wings and halteres.

**Figure supplement 2.** The positions of wing (blue) and haltere (red) showing their relative phase for a representative fly with intact wings (**A**) and wings cut (**B**).

**Figure 3—video 1.** Left and top views of the soldier fly with a severed left wing-haltere link (sub-epimeral ridge).

https://elifesciences.org/articles/53824/figures#fig3video1

wingbeat frequency. Alternatively, a lesioned ridge could mechanically disrupt frequencies by acting as a free end that dissipates energy, thereby disrupting the overall resonant mechanics of the thorax.

To test for these possibilities, we first lesioned the ridges on both sides. In these flies, the increase in wingbeat frequency was even further restricted (<40 Hz). In another set of flies, we kept both the sub-epimeral ridges intact but ablated both halteres. Ablating halteres alters their feedback but does not mechanically disrupt the thoracic linkage network. Clipping wings of the haltere-ablated flies resulted in significantly elevated wingbeat frequency (by ~90 Hz, *Figure 4*) as compared with flies with both sub-epimeral ridges lesioned ($p < 0.05$) but similar to flies with an intact sub-epimeral ridge. This suggests that mechanical integrity of these linkages most likely determines wingbeat frequency. Hence, the mechanical integrity of the entire thoracic linkage system, including both the scutellum and sub-epimeral ridge, is essential to maintain the resonant properties of the thorax.

## The effect of sub-epimeral ridge is unidirectional

If the sub-epimeral ridge is bidirectional, then wing frequency should change when haltere frequency is experimentally altered, and vice versa. Halteres, like wings, are powered by myogenic musculature, and thus changing the haltere mass affects its frequency. To test the hypothesis that sub-epimeral linkage is bidirectional, we loaded halteres with small weights, thereby altering their frequency, and measured the effect on wingbeat frequency. Unlike the wing clipping, we could not reduce the haltere mass in discrete steps as it is mostly concentrated at the end knob. Instead, we sequentially loaded each haltere knob with small amounts of glue, thus reducing its frequency in discrete steps (Section 1,2; *Figure 5—figure supplement 1A–C*). Small loads did not affect haltere frequency, but as the load increased beyond a threshold, haltere frequency decreased in discrete steps. However, wingbeat frequency remained constant in these experiments (*Figure 5*; for individual fly data, see *Figure 5—figure supplement 2A–F*, $p = 0.014$; one-sided Wilcoxon signed-rank test for left loaded haltere and wing pair at load2 and load3, n = 6 flies). Thus, the effect of sub-epimeral ridge is a unidirectional, which couples haltere motion to wing motion but not vice versa. To rule out the possibility that loading the haltere irreversibly damaged haltere muscles or the sub-epimeral ridge, we detached the load and confirmed that haltere frequency recovered and again matched wingbeat frequency (*Figure 5*, *Figure 5—figure supplement 2*).

In addition to disrupting the coordination between wing and halteres, a perturbation to the haltere motion would also alter the haltere feedback to the wing. To test for the effect of aberrant haltere feedback on wing kinematics, we quantified the wing amplitude for two of our treatments – flies with left-side sub-epimeral ridge lesioned as well as flies with loaded halteres. In the former case, the lesioned link disrupts the phase coordination, thereby altering the feedback at all wing lengths. In flies with loaded halteres, the added load on the haltere may alter its feedback even at values for which the wing-haltere are perfectly coordinated in phase and frequency. In both cases, we observed that the wing amplitude of the lesioned/loaded side was slightly reduced as compared with the intact side, although this effect is not consistent across the treatments (*Figure 5—figure supplement 4A and B*). For the left, sub-epimeral-lesioned flies, both the left and right wings reduce their amplitude as we clip the wings symmetrically ($p = 0.008$ and $0.024$ for left and right, respectively, Kruskal–Wallis H-test, *Figure 5—figure supplement 4A*) consistent with our previous finding on the effect of wing clipping on amplitude. However, the wing on the left lesioned side shows a greater reduction as compared with the right wing ($p < 0.05$ for left-right wing pairs at all wing lengths, two-sided Wilcoxon signed-rank test, n = 5). Additionally, for the treatment with loaded halteres, despite a slight decrease in wing amplitude on the loaded side with increase in the haltere load, this effect is not significant when we compare the left wing across increasing loads ($p > 0.05$, Kruskal–Wallis H-test, *Figure 5—figure*

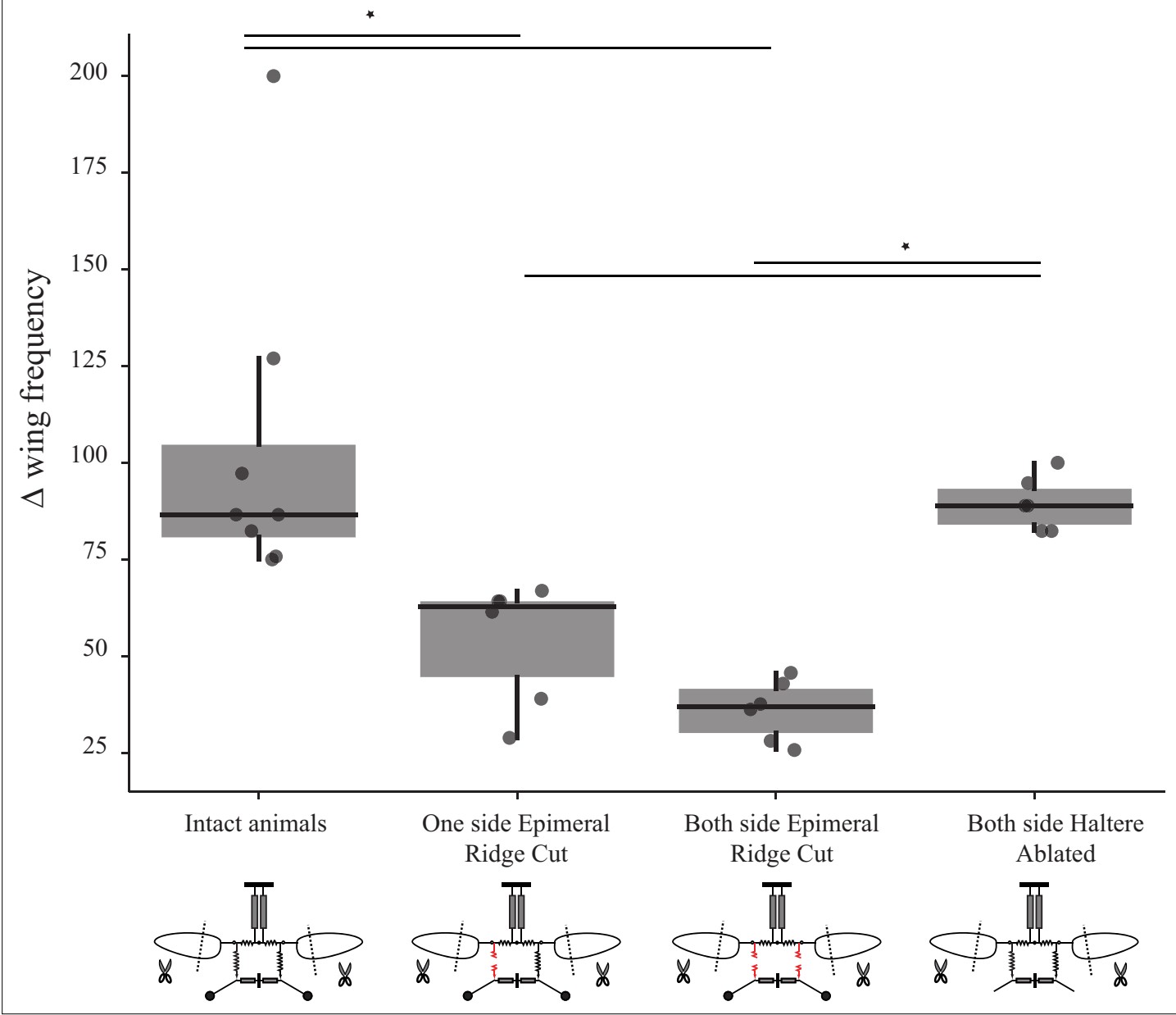

**Figure 4.** Sub-epimeral ridge lesions alter the resonant properties of the thorax. Individual box plots show the difference in wingbeat frequency between intact wing and at the shortest wing lengthin different treatment groups. Clipping wings of flies in which one or both sub-epimeral ridges are lesioned results in a smaller increase in wingbeat frequency as compared to flies with intact sub-epimeral ridges. In contrast, reducing wing length in flies in which both halteres are ablated increases wingbeat frequency significantly more than in flies with lesioned sub-epimeral ridges (p-value<0.05, non-parametric Kruskal–Wallis ANOVA followed by a Tukey–Kramer post hoc multi-comparison test, n; eight intact flies, six flies for the other three treatments). Insets under each group show the schematic for the four treatments: flies with an intact thorax, flies with one lesioned sub-epimeral ridge, flies with both sub-epimeral ridge lesioned and flies with both halteres ablated.

*supplement 4B*, n = 6) or the left-right pairs at each load (p>0.05, two-sided Wilcoxon signed-rank sum test, n = 6). Also, the wing amplitude on the loaded side is smaller than the intact side even when the load is removed.

## Discussion

For stable flight, bilateral wing motion must be precisely coordinated, and the halteres must maintain a precise phase difference relative to wings to ensure correct feedback to wing-steering muscles (*Deora*

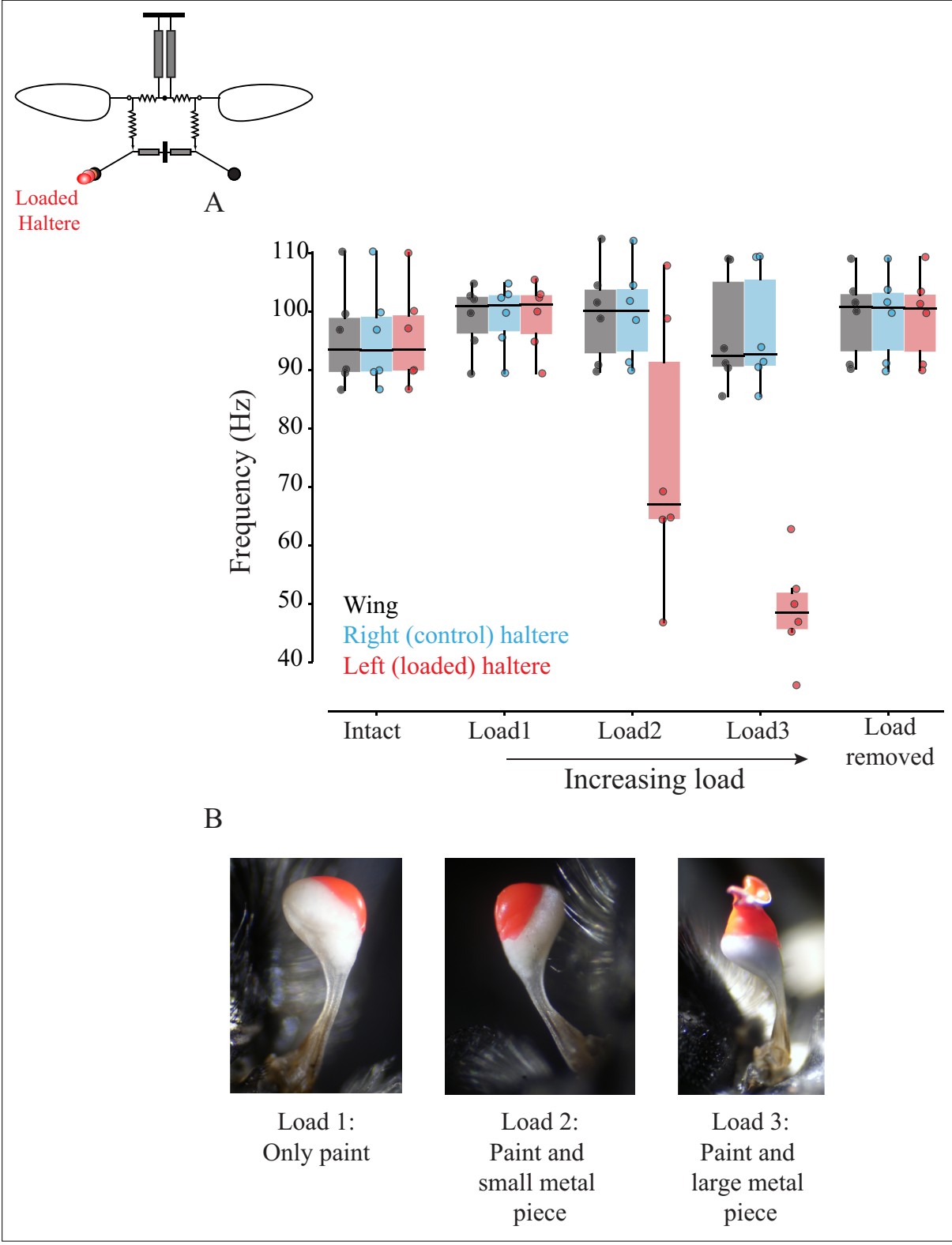

**Figure 5.** Wing-haltere linkage is unidirectional because changing haltere frequency does not alter wingbeat frequency. (**A**) Frequency of wing (gray), control haltere (blue), and loaded haltere (red) across all flies. Each dot represents a single individual fly. The haltere frequency drops as the haltere is loaded but the wingbeat frequency remains unaltered, showing that wing-haltere coupling is unidirectional. Inset shows the schematic for the treatment. Data for individual flies can be found in *Figure 5—figure supplement 2*. (**B**) Representative images of the haltere loaded with different amounts of load (also see *Figure 4*).

*Figure 5 continued on next page*

*Figure 5 continued*

The online version of this article includes the following video and figure supplement(s) for figure 5:

**Figure supplement 1.** Increasing amounts of glue, paint, and metal shavings were added to load halteres.

**Figure supplement 2.** Wing haltere coupling acts unidirectionally such that changing haltere frequency does not influence wing frequency.

**Figure supplement 3.** The wing (blue) and haltere (red) position showing their relative phase for a representative fly with intact halteres (**A**), loaded halteres (**B**), and the load removed (**C**).

**Figure supplement 4.** Aberrant haltere feedback causes reduction in wingbeat amplitude, although these results are not consistent across experiments.

**Figure 5—video 1.** Top view of soldier fly with left haltere loaded.

https://elifesciences.org/articles/53824/figures#fig5video1

*et al., 2015*; *Fayyazuddin and Dickinson, 1999*; *Fry et al., 2003*). Here, we show that the phase and frequency of wings and haltere motion are mechanically coupled by thoracic linkages, thereby imparting robustness of wing-wing and wing-haltere coordination against damage or wear-and-tear.

## Mechanical linkages enable robust frequency-phase output despite asymmetric wing damage

Physical damage to the wings of adult insects is irreversible and potentially deleterious for their fitness. Wings of certain insects have flexible costal break or specific venation patterns that prevent wing damage (*Mountcastle and Combes, 2014*). However, despite such adaptations, insects incur wing damage in the wild due to factors such as predation and age (*Cartar, 1992*; *Hayes and Wall, 2002*). When one or both wings are damaged, the overall aerodynamic load reduces, thereby increasing the frequency of thoracic and wing oscillations (*Deora et al., 2015*). The linkages described here ensure that the phase and frequency matching between wings and halteres is robust despite significant changes in wingbeat frequency, and hence they may be viewed as evolutionary adaptations that impart robustness to wing motion in case of damage or wear-and-tear.

Natural wing damage is typically asymmetric. Although shortening both wings symmetrically results in elevated wingbeat frequencies (*Deora et al., 2015*), shortening only one wing by as much as 50% did not significantly alter the resonant frequency of thorax. This suggests that the resonant frequency is dictated primarily by the wing with greater aerodynamic load. Importantly, in flies with an intact scutellar linkage, asymmetric changes in wing length did not alter the overall synchrony; wings remained bilaterally coordinated, regardless of their length (*Figure 2*).

Freely flying flies and bees with damaged wings increase their wingbeat frequency to compensate for the reduced lift (*Hedenström et al., 2001*; *Muijres et al., 2017*; *Vance and Roberts, 2014*). Moreover, flies and bees with asymmetric wing damage balance the roll torque by flapping the damaged wing at greater amplitudes (*Muijres et al., 2017*; *Vance and Roberts, 2014*). In our experiments with tethered flies, however, asymmetric wing damage in flies with intact thorax did not increase wingbeat frequency. Instead, in these flies, the clipped wing flapped at reduced wing amplitudes. These results suggest that the increase in wingbeat frequency and the amplitude modulation that is observed in freely flying insects with damaged wings may be an active response that enables compensation for the loss of the lift due to wing damage. Lesioning the scutellum completely decoupled the frequencies of the two wings of different lengths; shortened wings oscillated at frequencies up to twice that of the intact wing and cause irregular wing motion. These results underscore the importance of the scutellar linkage in ensuring precise coordination between the two wings.

## Control of bilateral kinematics by indirect flight muscles

In the experiments in which the scutellar link was lesioned, the shortened wing flapped at higher frequency than the intact wing. This shows that the indirect flight muscles on the two sides are, in principle, capable of operating at different frequencies, but are constrained to flap synchronously by the scutellar link. These results are consistent with the idea that indirect flight muscles aid the direct flight muscles in power modulation and kinematic control of wings (*Gordon and Dickinson, 2006*; *Lehmann et al., 2013*). In intact flies, power modulation occurs under constraints of equal bilateral wingbeat frequency and phase, which leaves open the possibility that amplitude or stroke plane can be modulated by power muscles. For instance, during turns, the two sets of dorsolongitudinal muscles

can be differentially activated by their motor neurons (*Gordon and Dickinson, 2006*; *Lehmann et al., 2013*; *Sponberg and Daniel, 2012*), potentially leading to differential power output. Moreover, asymmetric wing damage results in net torque on the body in hawkmoths (*Fernández et al., 2012*; also see, *Kihlström et al., 2021*). Asymmetric wing damage in hawkmoths causes activation delay in the dorsoventral power muscle, resulting in a voluntary yaw-like maneuver towards the undamaged side, perhaps to compensate for the reduced lift on the damaged side. It is important to note that although hawkmoths have indirect flight muscles, they are synchronous and hence under direct neural control. Our results suggest that, like hawkmoths, the indirect, asynchronous flight muscles of flies could also modulate power output of the ipsilateral wing, independent of the contralateral power muscles.

## Role of sub-epimeral ridge in wing-haltere coordination

The sub-epimeral ridge synchronizes the frequency and phase difference between wings and halteres on each side (*Figure 3*, *Deora et al., 2015*). How would the structural diversity of the thorax and linkages in diverse Diptera influence wing-haltere motion? For example, as compared with the rounded thorax with an almost circular sub-epimeral ridge in blowflies, the thorax of mosquitoes is thinner with an oblong sub-epimeral ridge. Presumably, there is also variation in the material properties and strain transfer across the cuticle. Across flies, there are also differences in their haltere kinematics. For example, blowfly halteres flap exactly antiphase to the wings, whereas, for mosquitoes, this phase difference is closer to 0.

It is not clear how the precise phase difference is set in these flies, but these parameters are likely the outcome of the variation in the thoracic and linkage anatomy across Diptera and the physics of coupled, driven oscillator systems. Indeed, when wing and haltere frequencies are decoupled by loading the haltere (*Figure 5*), the haltere oscillates at an altered phase relative to the wing, even when sub-epimeral ridge is intact (*Figure 3—figure supplement 2*, *Figure 5—figure supplement 3*). Their frequency, on the other hand, is thought to be determined primarily by the stretch-activation properties of their main driving muscles – the indirect flight muscles for wings, and haltere muscles (or Pringle's muscle) for halteres. Because the sub-epimeral ridge couples haltere frequency to wingbeat frequency, their vibration frequency is fine-tuned by this linkage and the overall thoracic geometry.

## Weak coupling properties of the sub-epimeral ridge

The sub-epimeral ridge weakly couples wings and halteres. Its stiffness is limited by its material strength and geometry, and it can ensure coordination of wing-haltere frequency close to original frequency. At wingbeat frequencies greater than about 150% of the original frequency, the haltere frequency reverts to a value that is closer to or equal to its natural frequency. This behavior is typical of independently driven oscillators, coupled by a mechanical element of finite coupling strength (*Strogatz, 2018*). Moreover, the sub-epimeral ridge acts in a unidirectional manner (*Figure 5*); moderate changes in wingbeat frequency alter the haltere frequency, but not vice versa. This may be the outcome of the large difference between the wing and haltere loads (and therefore inertia), and their respective muscles. Our experiments show that the aerodynamic load on the wings determines flapping frequency, and the halteres merely follow the wings.

## Role of haltere feedback in modulating wing amplitude

Aberrant haltere feedback due to a lesioned sub-epimeral ridge causes a reduction in the amplitude of the ipsilateral wings as compared to the contralateral (intact) side. For the flies with loaded halteres, we observe a similar although not significant trend of reduced wing amplitude on the ipsilateral-treated side (*Figure 5—figure supplement 4B*). *Prima facie*, this suggests that haltere feedback helps maintain or increase wing amplitude. However, these results must be interpreted with caution for several reasons. First, in all our experiments, flies are in an open-loop tether and do not need to offset their body weight by adjusting the lift forces generated by flapping wings. Hence, wing amplitude may be subject to greater variability. Second, under open-loop conditions, it is not clear how sensory feedback, via their role in steering muscle activation, influences wing amplitude. Third, although we lesion the sub-epimeral ridge or load halteres on one side, contralaterally projecting haltere interneurons (cHINs) continue to provide feedback (*Strausfeld and Seyan, 1985*). In addition, the large variation of wing amplitude in the haltere-loaded treatments – with and without load – can also be partially explained by the fact that flies with loaded halteres tend to groom their halteres more frequently

and may often block the wing motion by holding it at its the base (black arrows in *Figure 5—figure supplement 4B*). Future work is required to better understand the implications of thoracic biomechanics on haltere feedback and wing kinematics.

## General implications for other insects with asynchronous muscles

Although our study was focused on Dipteran insects, several implications of this study extend beyond Diptera. Indeed, all insects that possess asynchronous flight muscles must rely on linkages for wing-wing coordination. These include insects of the orders Hymenoptera (e.g., wasps, honeybees, etc.) and Coleoptera (e.g., beetles), which are hyper-diverse. In these insects, as in Diptera (which are also hyper-diverse), the bilateral coordination of wings is unlikely to be achieved through direct neural control as they flap at very high wingbeat frequencies. We therefore predict that similar linkages also exist in these insect taxa. In such insects, it is also likely that relative coordination between the front and the hindwings is mediated by passive linkages analogous to the sub-epimeral ridge in Diptera. Thoracic linkages may also play a very important role in miniature insects, in which there are fewer muscle groups for higher-level control. Extreme miniaturization in insects of the order Diptera, Hymenoptera, and Coleoptera (*Polilov, 2015*) poses severe physiological and biomechanical constraints on the organism (*Polilov, 2012*; *Sane, 2016*). Because smaller wings generate reduced aerodynamic lift, miniature insects must flap at increased frequencies to generate enough lift to stay aloft. In such insects, thoracic linkages are likely to play a key role in mediating synchronization of the wings, while also constraining their degrees of freedom. Small body sizes are also associated with rapid wingbeat frequencies, often at rates much greater than possible through direct neural control (*Dudley, 2000*). The key results in this paper show that the thoracic morphology of the fly plays an important role in providing robust wing coordination despite wing damage. How thoracic morphology and material characteristics are adapted for such rapid, coordinated wing movements remains a fascinating question for future studies.

## Materials and methods
### 1.1 Soldier fly rearing procedure
Wild-caught black soldier flies, *H. illucens,* were enclosed in vials filled with a medium of corn flour and agar mixed with yeast powder. In most cases, we caught wild gravid females that laid eggs immediately upon capture. The larvae were reared on the artificial medium. Adult soldier flies were maintained in mesh cages on a 12:12 hr day-night cycle. Some flies were reared on compost in which wild females laid eggs and the larvae fed and pupated. Pupae were collected in a separate box. Adults were maintained in natural day-night cycle. Animals reared in this manner were typically bigger and more active than lab-reared animals but showed no difference in behavior.

### 1.2 Surgical treatments and tethering procedure
1–4-day-old soldier flies were cold anesthetized by placing them in an ice box for 5 min. We performed surgical treatments (see *Figure 1C*) before their recovery from cold anesthesia. (a) In *control flies*, no surgeries were performed, but these insects were handled similar to experimental flies. (b) In *scutellum-lesioned flies,* we made a small cut only in the scutellum using a scalpel blade (#11, Fine Science Tools Inc, Foster City, CA) while leaving the rest of the thorax intact (left panel, *Figure 1C*). (c) In *unilateral sub-epimeral ridge-lesioned flies,* we lesioned the sub-epimeral ridge at a position anterior to the spiracle on the left side of the thorax (middle panel, *Figure 1C*). The right ridge was kept intact and served as internal control. To examine how sub-epimeral ridge influences wingbeat frequency, we compared data from *unilateral sub-epimeral-lesioned* group with previously published data on *control* flies (*Deora et al., 2015*). In both cases, procedures for rearing, handling, and tethering were identical. (d) In *bilateral sub-epimeral ridge-lesioned flies,* we lesioned sub-epimeral ridges on both sides. (e) In *bilateral haltere-ablated flies*, we cut out the knob of halteres on both sides.

Post surgery, we tethered the insects with cyanoacrylate adhesive to the tip of a needle bent to ~90° while they were kept on a pre-chilled metal block. The bent tip provided the necessary surface area to glue a fly to the tether. The tether was lowered using a three-way micro-manipulator (Narishige Scientific Instrument Laboratory, Tokyo, Japan) and attached dorsally on the fly scutum (anterior region of dorsal thorax). Flies were given at least an hour for recovery before recording their flight.

## 1.3 Wing-haltere perturbations and filming procedure

We positioned flies at about 60° to the horizontal (approximately its position during free flight) and elicited flight by lightly touching their abdomen with a brush. We used three high-speed cameras (v7.3 Phantom camera, Vision Research Inc, Wayne, NJ) to film the insects in flight at a resolution of 800 × 600 pixels at 2000 frames per second (~15–20 times the wingbeat frequency) and 100 μs exposure. Three cameras (one top view and two side views) captured the 3D motion of both pairs of wings and halteres (*Figure 1A*). The three camera views were calibrated for each filming bout using a custom-made calibration object.

### Flies with reduced wing length

We first filmed the flight of flies with intact wings, and then switched off the lights to inhibit their flight. Under a dissection microscope attached with a light source and a red-filter to cut off wavelengths below 610 nm, we clipped their wings with a pair of scissors (Fine Science Tools Inc) to an appropriate length using wing vein patterns as landmarks. To test the coupled dual-oscillator model (*Figure 1B*), we clipped one wing (either left or right) while keeping the other intact (*Figure 1C*). In all other experiments, both wings were symmetrically reduced. After each round of wing clipping, we filmed the experimental fly in flight. Each series of experiments yielded 4–6 data points, including 1 intact and 3–5 reduced wing lengths.

### Flies with loaded halteres

We initially filmed flies with intact wings and halteres, which served as control. Next, we loaded the left haltere with varying amounts of glue and metal shavings under a dissection microscope in red light. Because of the small size of halteres, the amount of load could not be accurately quantified, but we systematically decreased the haltere frequency by incrementing an arbitrary amount of glue and load mixture (right panel, *Figure 1C*). First, we loaded the haltere with glue (Fevicol; polyvinyl acetate, Pidilite Industries) mixed with red poster paint using a metal insect pin. We next added increasing amounts of aluminum shavings mixed with glue and paint to the already loaded haltere. Following each load increment, we filmed the tethered fly. After three rounds of loading, we carefully removed the film of glue, paint, and metal shaving with a pair of forceps under the microscope and again filmed the flight recovery. In a few cases, flies removed the load while self-grooming. This experimental procedure yielded five data points for each fly – one intact, three increasing amounts of load, and one with the load removed.

## 1.4 Analysis of videos

To compute the time period and frequency of a single wing and haltere stroke, we counted the number of frames per wingstroke from the videos. We analyzed 20 strokes per flight bout at each wing length value. In the *scutellum-lesioned group,* the amplitude and frequency of flies often varied, making it difficult to ascertain the precise time duration of each wing stroke. We digitized videos of these animals using the DLTdv3 code (*Hedrick, 2008*) in MATLAB (MathWorks Inc, Natick, MA) and analyzed the data using custom MATLAB codes. For each wing, we calculated the azimuthal (theta) angle, plotted the theta position of both wings as a function of time, and obtained the average wingbeat frequency using Fourier analysis. We calculated the change in wingbeat frequency after clipping wing length (ΔF) as

$$\Delta F = F_s - F_i \tag{1}$$

where $F_s$ is the wingbeat frequency at the shortest wing length and $F_i$ is the wingbeat frequency at the intact wing length. Increment in wingbeat frequency due to reduced wing length (ΔF) for all four groups was compared using non-parametric Kruskal–Wallis ANOVA followed by the Tukey–Kramer post hoc multi-comparison analysis.

Using DLTdv8 code in MATLAB (*Hedrick, 2008*, MathWorks), we digitized three calibrated camera views to track the position of tips and bases of both wings and halteres and the head-thorax joint. We used the deep learning module of the above software to digitize the wing tips. All other body parts were digitized either manually or with a cross-correlation tracker. The neural network was trained for 525 iterations in 20 epochs until the weights converged. Errors in automated tracking were manually

corrected across all videos. To estimate digitizing errors, we computed the average variation in wing and haltere length. Across all videos, there was an average variation of 1.95 and 6.97% in wing and haltere length, respectively. For amplitude analysis, we first transformed all the digitized points to a fly coordinate frame of reference, and then computed the wing (and haltere) amplitude using the angle made by the projection of the wing (and haltere) vector onto the average stroke plane and the pitch axis for each wingbeat. We analyzed ~20 wing/haltere stroke and computed the mean amplitude for each video. All data were compared using either Wilcoxon signed-rank test for comparison across pairs or Kruskal–Wallis H-test for comparing more than two groups. Codes used for all analysis can be found on GitHub, (copy archived at swh:1:rev:b63ccbcf8571d78e1317796c9a4c3fc80b466905, *Deora, 2021*).

## Acknowledgements

We thank Prof. Sandeep Krishna, Prof. Rudra Pratap, Rizwana Parween, and Akash Vardhan for the discussions on mechanical oscillators, Dr. Anand Krishnan and Dr Hardik Gala for discussions and suggestions on the manuscript and figures, and Abin Ghosh and Umesh Mohan for helping with training SSS on digitizing software and the NCBS IT team for streamlining data transfer during the pandemic shutdown.

## Additional information

### Funding

| Funder | Grant reference number | Author |
| --- | --- | --- |
| Air Force Office of Scientific Research | FA2386-11-1-4057 and FA9550-16-1-0155 | Sanjay P Sane |
| Human Frontier Science Program | | Tanvi Deora |
| Tata Institute of Fundamental Research | | Sanjay P Sane |
| Department of Science and Technology | | Sanjay P Sane |

The funders had no role in study design, data collection and interpretation, or the decision to submit the work for publication.

### Author contributions

Tanvi Deora, Conceptualization, Data curation, Formal analysis, Investigation, Methodology, Software, Validation, Visualization, Writing – original draft, Writing – review and editing; Siddharth S Sane, Digitized videos, Formal analysis, Writing – review and editing; Sanjay P Sane, Conceptualization, Funding acquisition, Investigation, Project administration, Resources, Supervision, Writing – original draft, Writing – review and editing

### Author ORCIDs

Tanvi Deora (ID) http://orcid.org/0000-0002-4019-2882
Sanjay P Sane (ID) http://orcid.org/0000-0002-8274-1181

### Decision letter and Author response

Decision letter https://doi.org/10.7554/eLife.53824.sa1
Author response https://doi.org/10.7554/eLife.53824.sa2

## Additional files

### Supplementary files

• Supplementary file 1. Tables of p-values for one-sided Wilcoxon signed-rank sum tests. (A) Table of p-values for a one-sided Wilcoxon signed-rank sum test at all wing length bins for epimeral ridge

cut flies in *Figure 3A*. (B) Table of p-values for a one-sided Wilcoxon signed-rank sum test between haltere and wing pairs at all haltere loads in *Figure 5*.

- Transparent reporting form

### Data availability

All data can be accessed on Dryad.

The following dataset was generated:

| Author(s) | Year | Dataset title | Dataset URL | Database and Identifier |
|---|---|---|---|---|
| Deora T, Sane SS, Sane SP | 2021 | Wings and halteres act as coupled dual-oscillators in flies | https://doi.org/10.5061/dryad.hqbzkh1cc | Dryad Digital Repository, 10.5061/dryad.hqbzkh1cc |

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

## Appendix 1

### Testing significance and estimating sample sizes

To estimate a minimum sample size to detect a difference between two groups, we used the estimated mean and SD of these two groups (measured from data) and simulated datasets of different sample sizes. For each simulated dataset, we calculated the p-value using the relevant test (Wilcoxon signed-rank test for all paired data and Kruskal–Wallis for groups). We used a bootstrapping method; repeating this about 10,000 times for each sample size and estimated the power, that is, probability of detecting a difference between these two groups at significance level of 0.05. Below we report the tested groups and power analysis for each experiment.

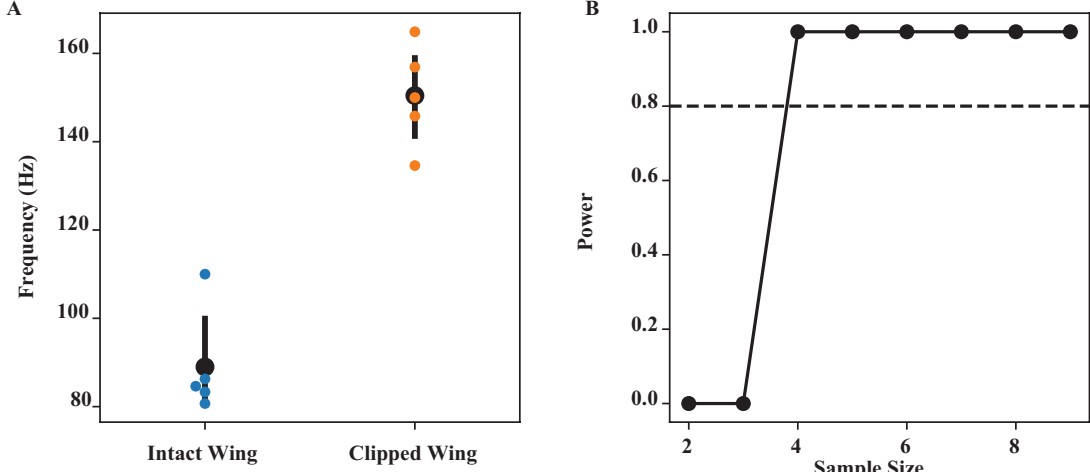

**Appendix 1—figure 1.** Wing coordination by scutellum (*Figure 2*). (**A**) The frequency of intact and clipped wing (at the wing length that has the largest difference) is significantly different (Wilcoxon signed-rank test, p=0.0215). (**B**) Power analysis for different sample sizes. Our sample size of 5 is greater than the minimum sample size (= 4) needed to have 80% confidence level (dashed line).

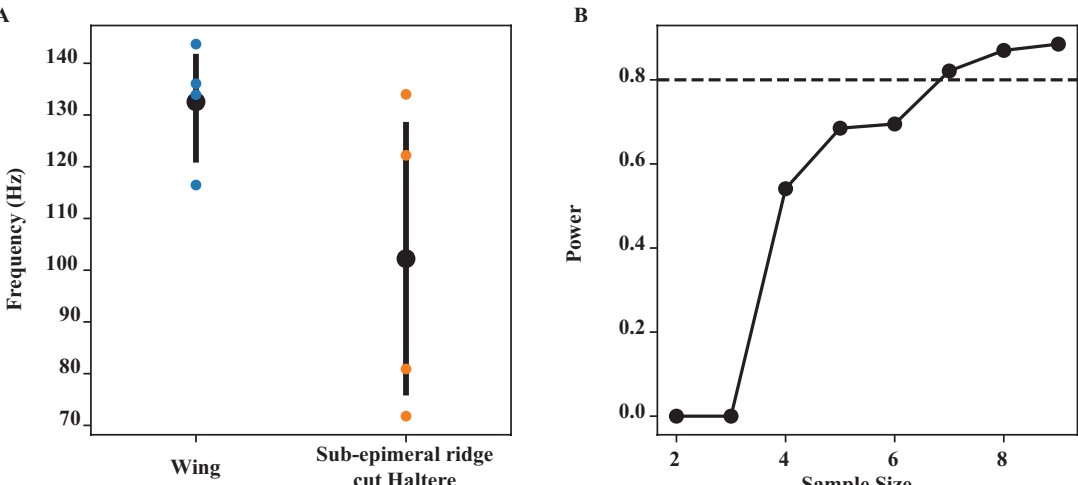

**Appendix 1—figure 2.** Wing-haltere coordination by sub-epimeral ridge (*Figure 3*). (**A**) The frequency of wing and epimeral ridge cut haltere at wing length bin of (06–0.7), which has the largest difference, is significantly different (Wilcoxon signed-rank test, p=0.039). (**B**) Power analysis for different sample sizes. At our current sample size of 5 at that wing length, we have a power of 0.693, that is, we have a 69.3% chance of picking up a significant difference.

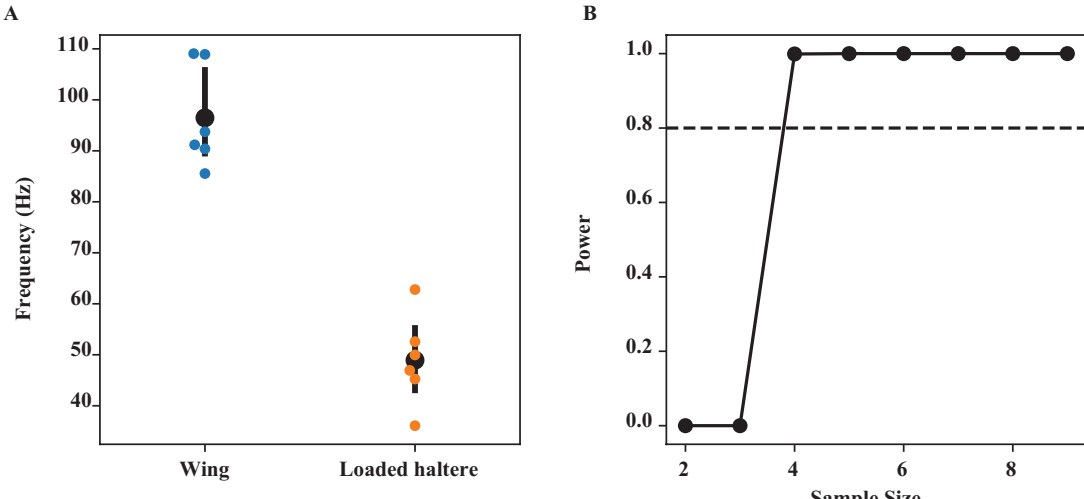

**Appendix 1—figure 3.** Loading haltere (*Figure 5*). (**A**) The frequency of wing and loaded haltere at maximum loading ('load3') is significantly different (Wilcoxon signed-rank test, p=0.014). (**B**) Power analysis for different sample sizes. Our sample size of 6 is greater than the minimum sample size (= 4) needed to have 80% confidence level (dashed line).

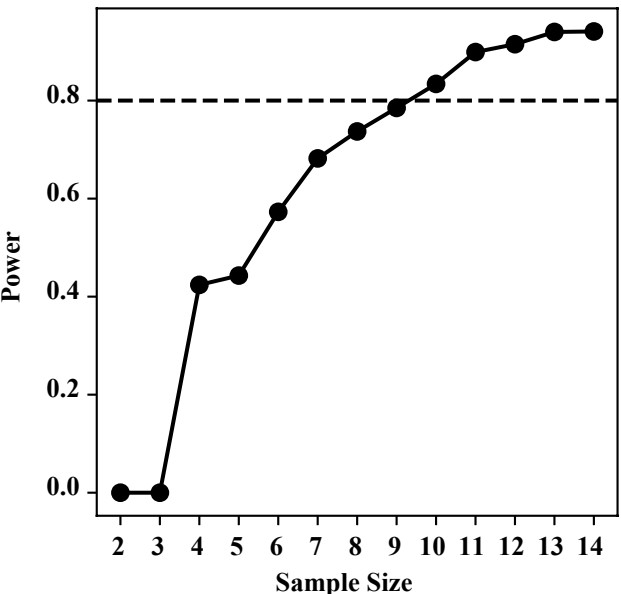

**Appendix 1—figure 4.** Integrity of sub-epimeral ridge is essential for thoracic resonance (*Figure 4*). For one-way Kruskal–Wallis test, our sample size n = 8 for control and 6 for the three treatments groups each. We used a similar bootstrapping method, simulating our data based on our group mean and SD, and calculating the power at difference sample sizes. To detect a significant difference at 80% chance, we require a minimum of nine samples. With our current sample size (n = 6), we have a substantial type II error: that is, a 42% chance of not detecting a difference if they were indeed different.

