## [Editor Report]

This manuscript examines how the mechanical linkages in the thorax of flies help these animals maintain symmetric wing motion in the face of uni- or bilateral wing damage. In previous work, the authors showed that these same linkages play an important role in maintaining the proper relative phase relationship between the wing and the haltere, a multifunctional sensory unique to flies. Through delicate manipulations of the thorax, wing, and haltere, the authors' experimental results support a mechanical model of the thorax they previously proposed known as the coupled dual-oscillator hypothesis, where mechanical linkages in the thorax both enable symmetric wing motion as well as coordinate haltere oscillation relative to the wing.

---

## [Decision Letter]

**Decision letter after peer review:**

Thank you for submitting your article "The coupled dual-oscillator model of wing and haltere motion in flies" for consideration by *eLife*. Your article has been reviewed by 3 peer reviewers, and the evaluation has been overseen by a Reviewing Editor and Aleksandra Walczak as the Senior Editor.

The reviewers have discussed the reviews with one another and the Reviewing Editor has drafted this decision to help you prepare a revised submission.

We would like to draw your attention to changes in our policy on revisions we have made in response to COVID-19 (https://elifesciences.org/articles/57162). Specifically, when editors judge that a submitted work as a whole belongs in eLife but that some conclusions require a modest amount of additional new data, as they do with your paper, we are asking that the manuscript be revised to either limit claims to those supported by data in hand, or to explicitly state that the relevant conclusions require additional supporting data.

Our expectation is that the authors will eventually carry out the additional experiments and report on how they affect the relevant conclusions either in a preprint on bioRxiv or medRxiv, or if appropriate, as a Research Advance in eLife, either of which would be linked to the original paper.

Summary:

The manuscript studies the mechanical wing-wing and wing-haltere links in diptera. The reported experiments, which have been performed on soldier flies, include lesions of specific thoracic components, clipping the wings and loading the halteres with extra mass. Wing and haltere frequencies have been measured using fast imaging. The main results show that the scutellum is central in wing-wing coupling and that the sub-epimeral ridge is important in the wing-haltere coupling.

Essential revisions:

1. The coupled dual-oscillator hypothesis was proposed in a previous paper from the authors five years ago. As a result, the findings in this paper come across as elaborations on the main points of that paper, rather than new, deeper insights. This could be helped by more fully detailing the model and its predictions in the introduction of the paper. For example, why is it that the only kinematic variable examined throughout the paper is wing/haltere beat frequency? Stroke amplitude/deviation of both the wing and haltere could be altered in some way by the experimental treatments. Given the experimental setup and supplementary figures 6 and 7, it seems that these data should exist somewhere. Additionally, there are several references to the idea that changing the resonant frequency of one wing will affect its contralateral partner. Although we can see that adding or removing mass will alter the resonant frequency of the whole system, it is unclear to me how removing wing mass will increase the resonant frequency of the intact wing to match the damaged wing. Again, I think more fully explaining the original model in the introduction will set up the results in a way that is easier to follow.

2. The initial framing of the paper was a bit confused. It starts with detailing how flies can accomplish amazing aerial manoeuvres that rely on "precise and rapid control" (ln 50) and emphasizes the role of mechanosensory feedback from the halteres. From there, it moves on to emphasize the importance of coordinating wing and haltere motion, and then thoracic functional morphology. One major issue with this section is that the idea of the haltere serving solely as a gyroscopic sensor is out of date. Dickerson et al. recently showed that the motion of these organs is under visual control, making them multifunctional sensory organs. More importantly, this initial framing that highlights the role of mechanosensory feedback gives the reader an impression that this paper will unite the study of the thorax's functional morphology with the biomechanics of haltere sensing. There is one place where these issues meet: examining how a lesioned sub-epimeral ridge impacts wing-haltere coupling. The authors put forth the hypothesis that such a lesion would lead disrupt haltere feedback to the wing steering system and alter wingbeat frequency. Whereas we are convinced that lesioning the ridge disrupts the resonant mechanics of the thorax, what is happening to the wing steering system is still a loose end. The proper experiment would be to lesion the sub-epimeral ridge and record from the wing steering muscle B1 and monitor its phase before and after the ridge is lesioned. If B1 fires earlier or later in the stroke cycle, that would demonstrate that these mechanical linkages play an important role in the neuromechanics of the steering system. Failing that, previous work (Tu and Dickinson, 1996; Heide and Götz, 1996) showed that B1 phase advances are associated with increases in wingstroke amplitude. Given the 3D data the authors already have, comparing wingstroke amplitude among flies where the sub-epimeral ridge is ablated would at least be a step toward testing both hypotheses. It would also link the biomechanics to muscle physiology and behavior.

3. One of the first outcomes of the results, on line 188, states that "the overall wingbeat frequency is determined by the frequency of the intact wing." However, there is no discussion of any changes in wing amplitude with clipping. Average load balancing between the left and right wings could still be achieved with a higher amplitude on the cut-wing side. Was there any change in amplitude?

4. Additionally considering the claim on line 188: There is no discussion of the role of the wing "gear box" in potential load balancing between the clipped and unclipped wings. As the wing is oscillated through a mechanical transmission then the gear ratio of this transmission will have a large role in the reflected inertia and drag force that the flight muscle must overcome. Does wing cutting result in changes in gear ratio? We are not asking for more experiments but rather we think this should be part of the discussion of potential phenomena for left-right load balancing (and thus unchanged wingbeat frequency).

5. Regarding the sentences beginning at line 296: I think the comparison between asymmetric clipping in bees, and the tethered experiments reported here need some further contextualization. The difference between tethered behavior and free flight behavior might be very different because of the need to generate aerodynamic load support in free flight. Further I am not sure how the claim "indicating that insects actively maintain their wingbeat frequencies" emerges from this comparison. There are other examples of wingbeat frequency modulation during tracking behaviors, such as Lehmann and Dickinson, 2001, that suggest (albeit) small changes in wingbeat frequency during vertical stripe tracking.

6. Line 291, the claim that "shortening the wing by as much as 50% did not significantly alter resonant frequency of the thorax" needs further support. Decreasing the wing length by half will have a large reduction on aerodynamic torque and inertial torque on the wing hinge. This is evident when the scutellum is cut and the clipped wing increases in frequency. When the scutellum is intact the frequency change is small. Can you provide some estimate for how much the combined wing loading will change when asymmetric clipping occurs?

7. The manuscript refers to differences in amplitude and phase of the wings and halteres (e.g lines 194, 203-204, 277, 287 and 294-295), yet the data provided contains almost exclusively frequency data. Reporting an oscillatory phenomenon by discussing only the frequency might be incomplete. The authors' previous paper (2015) on the same experimental system included phase measurements and several movies from which amplitude and phase coordination impairment following scutellum lesion was very clear. Here, because the amplitude and phase data (and movies) are not provided (excluding data samples in figures S6, S7), it is impossible to assess their true importance in this particular case. If these data are relevant (as the text might suggest), it would be advisable that the authors include relevant phase and amplitude data for the experimental treatments studied in the current manuscript. The lack of phase *data* is evident also in the discussion: line 277 "Here, we show that the *phase* and frequency of wings and haltere motion is mechanically coupled by thoracic linkages", as well as line 287.

8. The manuscript will gain a lot if supplementary movies are added (as in the authors' PNAS paper from 2015).

9. Results page 9, lines 233-254. Figure 4 shows the change in wingbeat frequency when both wings have been clipped under various lesions of the sub-epimeral ridge. Could the authors please elaborate on the wing clipping treatment? Specifically, from the schematics in figure 4 it implies that several wing lengths have been tested. As mentioned in the manuscript, the smaller the wing is, the higher its frequency (in the symmetric clipping), which makes one expect a wide distribution of frequencies when looking on the full range of wing clippings. However, figure 4 shows quite a narrow distribution of wingbeat frequencies in each column.

10. Results page 9, lines 233-254. The inference in this section relies on the notion that "ablating (the) halteres alters their feedback but does not mechanically disrupt the thoracic linkage network". Although it is obvious that ablated halteres cannot measure body rotations because their readout relies on the inertial Coriolis effect, it is not so obvious (to me) that their feedback is so much different in a tethered preparation, when there are no body rotations. If the haltere feedback (e.g. in terms of measuring body rotations) is irrelevant in a tethered prep, then it should not be surprising that the leftmost and rightmost columns of figure 4 show the same effect. Could the authors please refer to this point?

11. Results page 9, lines 256-271. The authors increase the mass of the haltere and measure a decrease in its beating frequency. The fact that in these experiments wingbeat frequency did not change is attributed to the claim that the sub-epimeral ridge is a mechanically unidirectional link. In the discussion the authors go back to this point saying that this "unidirectionality… may be the outcome of the large difference in the wing and haltere masses". I mostly agree with the latter point: the haltere mass is significantly smaller than the mass of the wing, hence, one should not expect that a change in the haltere frequency would have a significant mechanical effect on the wing beat frequency. Therefore, if the measurement is only due to an effect of very different masses, calling this a 'unidirectional link' might be an overstatement.

---

## [Author Response]

Essential revisions:1. The coupled dual-oscillator hypothesis was proposed in a previous paper from the authors five years ago. As a result, the findings in this paper come across as elaborations on the main points of that paper, rather than new, deeper insights. This could be helped by more fully detailing the model and its predictions in the introduction of the paper. For example, why is it that the only kinematic variable examined throughout the paper is wing/haltere beat frequency? Stroke amplitude/deviation of both the wing and haltere could be altered in some way by the experimental treatments. Given the experimental setup and supplementary figures 6 and 7, it seems that these data should exist somewhere. Additionally, there are several references to the idea that changing the resonant frequency of one wing will affect its contralateral partner. Although we can see that adding or removing mass will alter the resonant frequency of the whole system, it is unclear to me how removing wing mass will increase the resonant frequency of the intact wing to match the damaged wing. Again, I think more fully explaining the original model in the introduction will set up the results in a way that is easier to follow.

Thank you for your feedback. As the reviewers point out, this paper builds upon our earlier hypothesis of dual oscillator (Deora, Singh and Sane, 2015). The data in that paper suggested the hypothesis of the dual oscillator model, but this model is rigorously validated with the series of experiments that are reported here. We have now elaborated the predictions of coupled dual oscillator hypothesis (Results, subsection “Asymmetric wing damage influences kinematics but does not alter wing coordination”, first paragraph). In addition to providing the evidence for the coupled dual oscillator hypothesis, this paper provides several new insights especially in the context of how this system is adapted to withstand wing damage. Because we were focused on testing the model, the first version of our manuscript included only frequency data, which is determined by the resonance properties of the thorax. On the other hand, as explained in our general remarks, the amplitude data are difficult to interpret in the tethered scenario, in which both visual and haltere feedback (which mediate amplitude responses) do not provide an accurate representation of the flight status. Following the reviewer’s comments, we have now included amplitude analysis for various treatments.

Our main conclusions from the amplitude analysis are as follows:

a. Wing-wing coupling: When the wings of tethered flies with an intact thorax are clipped, they flap at a reduced amplitude as compared to flies with intact wings. This is contrary to expectations from freely-flying insects in which flies *increase* the amplitude of clipped wings to compensate for the loss in lift. Thus, the increased amplitude in clipped wings in freely flying insects is not the passive outcome of the resonant properties of the thorax; instead, it is actively controlled through feedback from various modalities. When sensory feedback is absent or aberrant (as in the case of the tethered flies in open loop), wing amplitude does not passively increase when their length is reduced.

When the wing-wing coupling element (scutellum) is lesioned, the wingbeat amplitudes are highly variable and show no consistent trend with reduced wing length. This is consistent with the role of scutellar lever arm in wing actuation. Lesioning the scutellum not only decouples the two wings but presumably also disrupts wing actuation, and hence amplitude control.

We have added these results to Figure 2C and D, in the Results, subsection “Amplitude control in response to wing damage” and Discussion, last paragraph of the subsection “Mechanical linkages enable robust frequency-phase output despite asymmetric wing damage”.

b. Role of haltere feedback in wing amplitude control: We examined the role of haltere feedback across two experiments. First, we lesioned the wing-haltere link (subepimeral ridge), which disrupted the relative phase and frequency coordination and caused aberrant mechanosensory feedback from haltere fields. Second, we loaded the haltere thereby disrupting the relative frequency coordination between wings and halteres, thereby again causing aberrant haltere feedback. Disruption in haltere feedback decreases amplitude on the ipsilateral side in tethered flight. It is however tricky to interpret these results (Figure 5 —figure supplement 4) any further because these insects are in an open-loop tether, and their wingbeat amplitude is not set by the dictates of steady flight. More importantly, haltere afferents provide feedback to both ipsi- and contra-lateral wings, *albeit* at different timescales.

2. The initial framing of the paper was a bit confused. It starts with detailing how flies can accomplish amazing aerial manoeuvres that rely on "precise and rapid control" (ln 50) and emphasizes the role of mechanosensory feedback from the halteres. From there, it moves on to emphasize the importance of coordinating wing and haltere motion, and then thoracic functional morphology. One major issue with this section is that the idea of the haltere serving solely as a gyroscopic sensor is out of date. Dickerson et al. recently showed that the motion of these organs is under visual control, making them multifunctional sensory organs. More importantly, this initial framing that highlights the role of mechanosensory feedback gives the reader an impression that this paper will unite the study of the thorax's functional morphology with the biomechanics of haltere sensing. There is one place where these issues meet: examining how a lesioned sub-epimeral ridge impacts wing-haltere coupling. The authors put forth the hypothesis that such a lesion would lead disrupt haltere feedback to the wing steering system and alter wingbeat frequency. Whereas we are convinced that lesioning the ridge disrupts the resonant mechanics of the thorax, what is happening to the wing steering system is still a loose end.

This is a valid point, and we have now cited Dickerson et al. in the Introduction, first paragraph. The focus of the paper is not as much on haltere sensory function, as it is on the oscillatory mechanics of the wing-thorax-haltere system. However, as the referees correctly point out, sensory function does factor into some of our results, such as in the experiments involving ablation of the sub-epimeral ridge. In general, any mention of halteres is incomplete without sufficient emphasis on its key sensory function as a balancing organ.

The proper experiment would be to lesion the sub-epimeral ridge and record from the wing steering muscle B1 and monitor its phase before and after the ridge is lesioned. If B1 fires earlier or later in the stroke cycle, that would demonstrate that these mechanical linkages play an important role in the neuromechanics of the steering system. Failing that, previous work (Tu and Dickinson, 1996; Heide and Götz, 1996) showed that B1 phase advances are associated with increases in wingstroke amplitude. Given the 3D data the authors already have, comparing wingstroke amplitude among flies where the sub-epimeral ridge is ablated would at least be a step toward testing both hypotheses. It would also link the biomechanics to muscle physiology and behavior.

As mentioned above, we have now added the data on wing stroke amplitude in flies with ablated sub-epimeral ridge. Lesioning the sub-epimeral ridge disrupts the relative phase of the haltere relative to the ipsilateral wing, suggesting that the wing ‘stumbles’ in its stroke cycle. Whether this is due to B1 misfiring, or some other steering muscle remains to be seen, but the most likely possibility is that this is due to aberrant haltere feedback. Although these data are useful to include, the additional insight is perhaps marginal.

3. One of the first outcomes of the results, on line 188, states that "the overall wingbeat frequency is determined by the frequency of the intact wing." However, there is no discussion of any changes in wing amplitude with clipping. Average load balancing between the left and right wings could still be achieved with a higher amplitude on the cut-wing side. Was there any change in amplitude?

We have now included the wingbeat amplitude data in Figure 2C and D, Results, subsection “Amplitude control in response to wing damage” and Discussion, last paragraph subsection “Mechanical linkages enable robust frequency-phase output despite asymmetric wing damage”. As also mentioned in the earlier comment, in flies with an intact thorax and asymmetric wing clipped, the wing beat amplitude of the clipped wing is lower than the intact wing. Previous work in various freely-flying insects report exactly the opposite: freely flying flies increase the amplitude of clipped wings to compensate for the loss in lift. Thus, our data suggest that wing amplitude is an actively controlled parameter (no surprises there) and when sensory feedback is in open loop, clipped wings do not passively increase amplitude when clipped.

4. Additionally considering the claim on line 188: There is no discussion of the role of the wing "gear box" in potential load balancing between the clipped and unclipped wings. As the wing is oscillated through a mechanical transmission then the gear ratio of this transmission will have a large role in the reflected inertia and drag force that the flight muscle must overcome. Does wing cutting result in changes in gear ratio? We are not asking for more experiments but rather we think this should be part of the discussion of potential phenomena for left-right load balancing (and thus unchanged wingbeat frequency).

We did not discuss the gear box in the context of the current paper, because we felt that it fell outside its scope. Also, we are beginning to question the gearbox hypothesis. We offer here a glimpse of our findings, which are being written up as a separate paper:

The “gear box” hypothesis was proposed by G Nalbach in her paper in 1984, and we – like others in the field – gathered (observational) evidence that seemed consistent with this hypothesis. More recently however, we (Sane lab) decided to directly test this hypothesis and investigate the role of the “gear box” in flight control by experimentally manipulating components of the gear box. In summary: our evidence is not consistent with this hypothesis. We offer the following lines of evidence: First, when we imaged the wing hinge of multiple species of flies, we found that not all of them possess the grooved structure of the pleural wing process (PWP) that is observed in some muscoids. Yet, they can all perform smooth turns and modulate their wing amplitude. Second, and most importantly, when we ablate the PWP in houseflies, they are capable of pitch and yaw maneuvers, both of which require wing modulation. These two lines of evidence suggest that the PWP- radial stop system may not function as a gear box quite in the way that we had previously thought. Instead of modulating wing kinematics by changing gear ratio of the mechanical transmission of forces, our results suggest that the gear box perhaps acts as a frictional surface that helps to limit the wing amplitude via contact with the radial stop (at the base of wing vein) helping the wing steering muscles that act as brake during the down stroke of the wing. These results again imply that wingbeat amplitude may be determined primarily by the wing steering muscles.

5. Regarding the sentences beginning at line 296: I think the comparison between asymmetric clipping in bees, and the tethered experiments reported here need some further contextualization. The difference between tethered behavior and free flight behavior might be very different because of the need to generate aerodynamic load support in free flight. Further I am not sure how the claim "indicating that insects actively maintain their wingbeat frequencies" emerges from this comparison. There are other examples of wingbeat frequency modulation during tracking behaviors, such as Lehmann and Dickinson, 2001, that suggest (albeit) small changes in wingbeat frequency during vertical stripe tracking.

We have now added some text (Discussion, last paragraph of subsection “Mechanical linkages enable robust frequency-phase output despite asymmetric wing damage” to elaborate on the comparison between bees and flies under both free flying and tethered conditions). The logic for active increase is also elaborated – it is quite simple: if the increase in amplitude is not the result of passive thoracic mechanics (as our results in Figure 2 show), then it must be driven actively.

6. Line 291, the claim that "shortening the wing by as much as 50% did not significantly alter resonant frequency of the thorax" needs further support. Decreasing the wing length by half will have a large reduction on aerodynamic torque and inertial torque on the wing hinge. This is evident when the scutellum is cut and the clipped wing increases in frequency. When the scutellum is intact the frequency change is small. Can you provide some estimate for how much the combined wing loading will change when asymmetric clipping occurs?

As all of our experiments were performed on tethered flies, wing loading (body weight / total area of the wings) is not a relevant parameter, as they do not offset their body weight. For this reason, we represented wing damage only as changes in wing length.

7. The manuscript refers to differences in amplitude and phase of the wings and halteres (e.g lines 194, 203-204, 277, 287 and 294-295), yet the data provided contains almost exclusively frequency data. Reporting an oscillatory phenomenon by discussing only the frequency might be incomplete. The authors' previous paper (2015) on the same experimental system included phase measurements and several movies from which amplitude and phase coordination impairment following scutellum lesion was very clear. Here, because the amplitude and phase data (and movies) are not provided (excluding data samples in figures S6, S7), it is impossible to assess their true importance in this particular case. If these data are relevant (as the text might suggest), it would be advisable that the authors include relevant phase and amplitude data for the experimental treatments studied in the current manuscript. The lack of phase *data* is evident also in the discussion: line 277 "Here, we show that the *phase* and frequency of wings and haltere motion is mechanically coupled by thoracic linkages", as well as line 287.

As described in our prefatory remarks, this comment took us the longest time to address because the amplitude data were not readily available and required us to digitize all the videos. We finally managed to do that and have now added amplitude and phase data across our treatment as described above.

8. The manuscript will gain a lot if supplementary movies are added (as in the authors' PNAS paper from 2015).

We have now included Supplementary movies for all our treatments.

9. Results page 9, lines 233-254. Figure 4 shows the change in wingbeat frequency when both wings have been clipped under various lesions of the sub-epimeral ridge. Could the authors please elaborate on the wing clipping treatment? Specifically, from the schematics in figure 4 it implies that several wing lengths have been tested. As mentioned in the manuscript, the smaller the wing is, the higher its frequency (in the symmetric clipping), which makes one expect a wide distribution of frequencies when looking on the full range of wing clippings. However, figure 4 shows quite a narrow distribution of wingbeat frequencies in each column.

Thank you for pointing this out. For figure 4 and section, we compare the change of frequency for the smallest wing length (= maximum wing damage) for all treatments. We have clarified this in the text on which now reads:

“In flies with unilaterally-lesioned sub-epimeral ridge, changes in wingbeat frequency were relatively moderate (~60 Hz) even after shortening the wing to the smallest wing length” We also specify this in the figure legend.

10. Results page 9, lines 233-254. The inference in this section relies on the notion that "ablating (the) halteres alters their feedback but does not mechanically disrupt the thoracic linkage network". Although it is obvious that ablated halteres cannot measure body rotations because their readout relies on the inertial Coriolis effect, it is not so obvious (to me) that their feedback is so much different in a tethered preparation, when there are no body rotations. If the haltere feedback (e.g. in terms of measuring body rotations) is irrelevant in a tethered prep, then it should not be surprising that the leftmost and rightmost columns of figure 4 show the same effect. Could the authors please refer to this point?

Our data show that when the wings are clipped, the resultant change in wingbeat frequency in insects with intact halteres is the same as change in wingbeat frequency with both halteres ablated. However, in insects with lesioned sub-epimeral ridges, the change in wingbeat frequency is less. Lesioning the epimeral ridge causes phase disruption between wing and halteres even at intact wing length (see Deora et al., 2015). Because haltere mechanosensors entrain to the phase of haltere beating, we suspect that the phase of firing relative to the wing is essential for haltere feedback. Lesioning the epimeral ridge would alter the phase, even in absence of body rotation, and hence affect wing kinematics.

This hypothesis is laid out in Results, second paragraph of the subsection “Integrity of the subepimeral right is essential for resonant oscillations of the thorax”.

11. Results page 9, lines 256-271. The authors increase the mass of the haltere and measure a decrease in its beating frequency. The fact that in these experiments wingbeat frequency did not change is attributed to the claim that the sub-epimeral ridge is a mechanically unidirectional link. In the discussion the authors go back to this point saying that this "unidirectionality… may be the outcome of the large difference in the wing and haltere masses". I mostly agree with the latter point: the haltere mass is significantly smaller than the mass of the wing, hence, one should not expect that a change in the haltere frequency would have a significant mechanical effect on the wing beat frequency. Therefore, if the measurement is only due to an effect of very different masses, calling this a 'unidirectional link' might be an overstatement.

We agree with this point. Nothing about link itself makes it unidirectional, but rather the effect is only due to the different masses.

We have changed the title of the subsection to “The effect of sub-epimeral ridge is unidirectional”. We also specify this in the Discussion, subsection “Weak coupling properties of the sub-epimeral ridge”.